# DUSP-1 Induced by PGE_2_ and PGE_1_ Attenuates IL-1β-Activated MAPK Signaling, Leading to Suppression of NGF Expression in Human Intervertebral Disc Cells

**DOI:** 10.3390/ijms23010371

**Published:** 2021-12-29

**Authors:** Takuya Kusakabe, Yasunobu Sawaji, Kenji Endo, Hidekazu Suzuki, Takamitsu Konishi, Asato Maekawa, Kazuma Murata, Kengo Yamamoto

**Affiliations:** Department of Orthopedic Surgery, Tokyo Medical University, 6-7-1 Nishi-Shinjuku, Shinjuku-ku, Tokyo 160-0023, Japan; takuyakskb0805@gmail.com (T.K.); kendo@tokyo-med.ac.jp (K.E.); hidekadu@hotmail.com (H.S.); taka.koni.0717@gmail.com (T.K.); asato.m19870510@gmail.com (A.M.); kaz.mur26@gmail.com (K.M.); kyamamoto@pf6.so-net.ne.jp (K.Y.)

**Keywords:** dual-specificity phosphatase (DUSP)-1, prostaglandin E_2_ (PGE_2_), prostaglandin E_1_ (PGE_1_), interleukin-1β (IL-1β), mitogen-activated protein kinase (MAPK), nerve growth factor (NGF), intervertebral disc (IVD)

## Abstract

The molecular mechanism of discogenic low back pain (LBP) involves nonphysiological nerve invasion into a degenerated intervertebral disc (IVD), induced by nerve growth factor (NGF). Selective cyclooxygenase (COX)-2 inhibitors are mainly used in the treatment of LBP, and act by suppressing the inflammatory mediator prostaglandin E_2_ (PGE_2_), which is induced by inflammatory stimuli, such as interleukin-1β (IL-1β). However, in our previous in vitro study using cultured human IVD cells, we demonstrated that the induction of NGF by IL-1β is augmented by a selective COX-2 inhibitor, and that PGE_2_ and PGE_1_ suppress NGF expression. Therefore, in this study, to elucidate the mechanism of NGF suppression by PGE_2_ and PGE_1_, we focused on mitogen-activated protein kinases (MAPKs) and its phosphatase, dual-specificity phosphatase (DUSP)-1. IL-1β-induced NGF expression was altered in human IVD cells by MAPK pathway inhibitors. PGE_2_ and PGE_1_ enhanced IL-1β-induced DUSP-1 expression, and suppressed the phosphorylation of MAPKs in human IVD cells. In DUSP-1 knockdown cells established using small interfering RNA, IL-1β-induced phosphorylation of MAPKs was enhanced and prolonged, and NGF expression was significantly enhanced. These results suggest that PGE_2_ and PGE_1_ suppress IL-1β-induced NGF expression by suppression of the MAPK signaling pathway, accompanied by increased DUSP-1 expression.

## 1. Introduction

The point prevalence of low back pain (LBP) in 2017 was estimated to be 7.8% of the global population, meaning that 577 million people were affected at any one time. The number of people with LBP has been increasing since 1990, mainly because of the ageing and increasing global population [1]. Discogenic LBP is one type of LBP, which is associated with intervertebral disc (IVD) pathology [2]. In a study of lumbar discography in patients with chronic LBP, 39% of the patients were reported to have LBP of IVD origin [3].

Nonphysiological nerve invasion into degenerated IVDs, owing to aging or inflammation, is considered as the molecular mechanism of discogenic LBP. Normal IVDs consist of the annulus fibrosus (AF) and nucleus pulposus (NP), and contain a large amount of extracellular matrix (ECM) components, such as aggrecan and type II collagen, which contribute to its unique avascular and aneural structure, except for the outer third of the AF. In other words, ECM components, particularly chondroitin sulfate of aggrecan, play an important role in protecting IVDs from peripheral nerve innervation [4,5]. The degeneration of IVDs occurs by the upregulation of proteinases, such as family of matrix metalloproteinases and a disintegrin and metalloproteinase with thrombospondin motifs, in response to inflammatory or mechanical stimulation [6,7]. Degeneration of the ECM within the IVD enables peripheral nerves to penetrate into the inner part of the NP, which also requires the upregulation of nerve growth factor (NGF) from stimulated IVD cells [2,8,9,10].

NGF is known to be a trophic growth factor for sympathetic and sensory nerve cells, which stimulates their differentiation, growth, maintenance, and survival [11]. On the other hand, several reports have indicated that NGF also has hyperalgesic properties [12,13,14,15,16]. The intensity of NGF staining is high in painful intervertebral discs, and a positive association between NGF expression and LBP has been reported [17]. In a recent clinical trial, tanezumab (a monoclonal antibody against NGF) was reported to provide long-term pain relief and functional improvement in patients with chronic LBP [18]. These findings suggest that NGF is deeply involved in the pathogenesis of discogenic LBP.

Recently, it has been clarified that persistent minor inflammation is involved in chronic LBP, including discogenic LBP, and induces NGF [19]. The proinflammatory cytokines interleukin (IL)-1β, IL-6, and tumor necrosis factor α are considered to be involved in LBP development, because they can induce a number of catabolic proteinases that cause IVD degeneration, as well as produce pain-associated molecules, including prostaglandin (PG) E_2_ and NGF in IVD cells [20,21,22,23]. Indeed, the neutralization of these cytokines using antibodies resulted in a reduction in the pathology of LBP [24,25,26]. Among these cytokines, IL-1β appears to most strongly induce NGF expression in human IVD cells [27,28,29], suggesting that IL-1β plays important roles as a master-trigger for inducing IVD pathogenesis.

In the clinical setting, nonsteroidal anti-inflammatory drugs (NSAIDs), including selective cyclooxygenase (COX)-2 inhibitors, have been primarily used for managing LBP [30]. These drugs are used to suppress the pain mediator PGE_2_, which is produced by COX-2 induced by inflammatory stimuli, such as IL-1β. However, our previous in vitro study using cultured human IVD cells demonstrated that the induction of NGF by IL-1β was suppressed by PGE_2_ or PGE_1_, and was augmented by a selective COX-2 inhibitor (NS-398) [31,32]. Hence, we suggested that the inhibition of PGE_2_ by a selective COX-2 inhibitor may have limited efficacy for LBP. However, the mechanism by which PGE_2_ and PGE_1_ suppress NGF expression remains unclear to date. Therefore, we focused on the mitogen-activated protein kinase (MAPK) pathway, which is the intracellular signaling pathway of inflammation, and the MAPK phosphatase dual-specificity phosphatase (DUSP)-1.

MAPKs are a family of protein kinases involved in the signal transduction of various extracellular stimuli, including IL-1β. They are activated via the phosphorylation of specific tyrosine and threonine residues. Extracellular signal-regulated kinase (ERK), c-Jun N-terminal kinase (JNK), and p38 are the typical MAPK classes in mammals [33]. MAPKs regulate not only inflammatory responses, but also various physiological processes, including cell proliferation, differentiation, apoptosis, and stress and immune responses [34,35,36,37].

DUSP-1 is a negative regulator of MAPK signaling, and dephosphorylates all three classes of MAPKs [38]. Among the 10 catalytically active DUSP family members, the regulatory role of DUSP-1, in particular, has been clarified in various cellular responses, particularly in inflammation [39,40]. DUSP-1 has been reported to play an important role in the development of arthritis. In addition, DUSP-1 expression in the synovium of patients with osteoarthritis was found to be lower than that in the synovium of normal subjects (GEO accession number GDS5401 and GDS5403) [41], and the incidence and severity of arthritis was significantly higher in DUSP-1-deficient mice than in wild-type mice [42]. However, to date, the role of DUSP-1 in human IVD and its molecular mechanisms remain unclear. Therefore, in this study, we focused on MAPKs and DUSP-1 in human IVD cells to elucidate the repressive mechanism of PGE_2_ and PGE_1_ in IL-1β-induced NGF expression.

## 2. Results

### 2.1. Effects of Inhibitors of Various MAPK Pathways on IL-1β-Induced NGF Expression in Human IVD Cells

We first investigated the involvement of MAPKs in IL-1β-induced NGF expression, using inhibitors of the various MAPK pathways. A p38 inhibitor strongly suppressed IL-1β-induced NGF expression in a concentration-dependent manner (Figure 1A), whereas NGF expression was increased by a MEK1/2 (an upstream kinase of ERK) inhibitor at high concentrations (5 and 10 μM) (Figure 1B), and was unchanged by a JNK inhibitor (Figure 1C). These results indicate that MAPK pathways are involved in IL-1β-induced NGF expression, but their mechanism of action may differ depending on the MAPK pathway.

### 2.2. PGE_2_ and PGE_1_ Suppressed IL-1β-Induced Phosphorylation of MAPKs in Human IVD Cells

We next investigated the regulation of IL-1β-induced phosphorylation of MAPKs by PGE_2_ and PGE_1_ in human IVD cells. p38, JNK, and ERK phosphorylation was detected as early as 15 to 30 min after IL-1β stimulation, and gradually decreased by 45 min. When cells pretreated with exogenous PGE_2_ or PGE_1_ for 3 h were stimulated with IL-1β, the phosphorylation of all MAPKs was suppressed at each time point compared with untreated cells (Figure 2).

### 2.3. PGE_2_ and PGE_1_ Enhanced DUSP-1 Expression in Human IVD Cells

We further investigated the regulation of DUSP-1 expression by PGE_2_ and PGE_1_ in human IVD cells. DUSP-1 expression increased with PGE_2_ or PGE_1_ treatment at 0.5 h, peaked at 1 h, and decreased at 3 h, but significantly high levels were maintained (Figure 3A). In addition, DUSP-1 expression was induced by IL-1β alone, and when PGE_2_ or PGE_1_ was present in the culture medium, DUSP-1 expression was enhanced in a concentration-dependent manner (Figure 3B). These results indicate that both PGE_2_ and PGE_1_ upregulate DUSP-1 expression in human IVD cells.

### 2.4. DUSP-1 Knockdown by Small Interfering RNA (siRNA) Transfection in Human IVD Cells

To investigate the role of DUSP-1 on IL-1β-induced NGF expression, DUSP-1 knockdown cells were prepared by a gene knockdown method using siRNA. First, we analyzed the transfection efficiency of the siRNA by transfecting human IVD cells with fluorescein isothiocyanate (FITC)-labeled siRNA, and found that more than 80% of the cells were positive for FITC (Figure 4A). Next, we confirmed that when DUSP-1-siRNA was transfected into human IVD cells, steady-state DUSP-1 mRNA was knocked down by more than 70% (Figure 4B).

### 2.5. IL-1β-Induced Phosphorylation of MAPKs Was Enhanced and Prolonged in DUSP-1 Knockdown Cells

We next compared the IL-1β-induced phosphorylation of MAPKs in untransfected cells and DUSP-1 knockdown cells by Western blotting. There was no significant difference in the phosphorylation of MAPKs 15 min after IL-1β treatment between the two types of cells. However, the phosphorylation of all MAPKs was increased in DUSP-1 knockdown cells at 30, 45, and 60 min after IL-1β treatment, and the increase was prolonged compared with untransfected cells. These results confirmed that DUSP-1 was functionally knocked down in our culture system (Figure 5).

### 2.6. IL-1β-Induced NGF Expression Was Enhanced in DUSP-1 Knockdown Cells

Finally, we compared IL-1β-induced NGF expression in untransfected cells and DUSP-1 knockdown cells. NGF expression was significantly enhanced in DUSP-1 knockdown cells compared with untransfected cells (Figure 6). These results indicate that DUSP-1 plays an important role in NGF expression.

## 3. Discussion

We previously reported that IL-1β-induced NGF expression is suppressed by PGE_2_ and PGE_1_ in human IVD cells [31,32], but its mechanism remained unclear. In the present study, we demonstrated that both PGE_2_ and PGE_1_ induce the expression of DUSP-1, which then suppresses the phosphorylation of MAPKs and NGF expression. To our knowledge, this is the first report that PGE_2_ and PGE_1_ induce DUSP-1 expression in human IVD cells.

MAPKs are closely involved in inflammatory diseases, such as rheumatoid arthritis, psoriasis, inflammatory bowel disease, neurodegenerative diseases, and cancer [43,44,45]. Hence, MAPKs have been considered important targets for therapeutic interventions for these diseases. However, to date, there have been no reports that MAPKs are involved in the regulation of NGF in IVD cells. In this study, we demonstrated that both PGE_2_ and PGE_1_ suppress IL-1β-induced phosphorylation of MAPKs. Among the MAPK pathway inhibitors tested, p38 showed the most robust suppression of NGF expression. This indicates that MAPK pathways are involved in NGF expression in human IVD cells, and p38 is likely to be a major pathway in the pathogenesis of discogenic LBP.

Inhibition of the p38 pathway with synthetic compounds (SB202190 and PD38059) has been shown to inhibit the inflammatory response and the catabolism of IVD cells, and to improve the stress-induced decrease in matrix anabolism [46,47]. In animal models, the use of p38 inhibitors in neuropathy, lumbar disc herniation, and lumbar canal stenosis (LCS) has been reported to improve pain and functional impairment [48,49], suggesting that p38 inhibitors have the potential to reduce neuropathic pain. These reports are consistent with our results that IL-1β-induced NGF expression was suppressed by p38 inhibition. However, to date, no p38 inhibitor has been established as a therapeutic agent. Although more than 20 types of p38 inhibitors have been developed to date for the possible treatment of other inflammatory diseases, only a few types have advanced to phase II clinical trials. Several MAPK inhibitors have been developed as possible anti-inflammatory agents, but their toxic side effects, owing to the roles of MAPK pathways in normal cellular physiology, have been a major obstacle [43,50].

On the other hand, DUSP-1 may be a new therapeutic target molecule because it widely suppresses the excessive phosphorylation of MAPKs. In this study, PGE_2_ and PGE_1_ were found to enhance IL-1β-induced DUSP-1 expression in human IVD cells. In addition, the induction of NGF expression was augmented in DUSP-1 knockdown cells, indicating that DUSP-1 is an important regulatory molecule for NGF expression in human IVD cells.

Although no reports to date have indicated that DUSP-1 may be useful for the treatment of LBP, some therapeutic agents for arthritis have been reported to induce DUSP-1. Steroid injection is effective for arthritis, and dexamethasone is known to enhance DUSP-1 expression [51,52]. Aurothiomalate, an antirheumatic drug, induces DUSP-1 expression in chondrocytes [53]. Hyaluronic acid, which is injected intra-articularly as a treatment for osteoarthritis, also induces DUSP-1 [54]. These results indicate that DUSP-1 may regulate arthritis, and the fact that IVD is mostly composed of chondrocytes suggests that DUSP-1 may also regulate discogenic LBP. The induction of DUSP-1 expression by PGE_2_ has not been reported to date, other than what we have shown previously in human synovial cells [55]. Therefore, the results of our present study, in which PGE_2_ and PGE_1_ induced DUSP-1 expression in human IVD cells, are considered to be highly novel.

In the treatment of discogenic LBP, PGE_2_ has been considered to be a mediator of inflammation and pain that should be suppressed. However, a selective COX-2 inhibitor (NS-398) suppressed PGE_2_-enhanced NGF expression in IL-1β-stimulated human IVD cells, and conversely suppressed NGF expression when PGE_2_ was extrinsically supplemented [31]. These results indicate that PGE_2_ is a mediator of inflammation and pain, but also has important bioactivity, namely, the inhibition of NGF expression in a negative feedback manner, and this bioactivity is mediated by DUSP-1.

In the present study, we showed that PGE_1_ induces DUSP-1 similarly to PGE_2_. We previously demonstrated that the PGE_1_ and PGE_1_ derivative limaprost suppresses NGF expression [32]. Because PGE_1_ shares plasma membrane receptors with PGE_2_, and their affinities are similar (Ki for E-series prostanoid receptor (EP) 2: 10 and 12 nM, respectively; and Ki for EP4: 1.9 and 2.1 nM, respectively) [56], we speculate that their induction of DUSP-1 and their inhibition of MAPK phosphorylation are comparable. Limaprost has been used for the treatment of LCS in Japan and some other Asian countries. It has been demonstrated to be effective in relieving LCS symptoms, including leg pain, leg numbness, and accompanying gait disturbance with intermittent claudication. Its mechanism of action has been thought to largely involve its property to increase peripheral blood flow by expanding blood vessels. Interestingly, limaprost has also been reported to relieve LBP [57,58]. It was suggested that limaprost may relieve LBP, in part by suppressing the phosphorylation of MAPKs and NGF expression by inducing DUSP-1.

This study has the following limitations. First, this study was an in vitro study, and the findings observed in this study may not completely reflect the in vivo situation. Second, it was difficult to detect the DUSP-1 protein with commonly available antibodies, and hence it was not possible to show changes in DUSP-1 at the protein level. Third, the IVD cells isolated by enzymatic digestion were a mixture of cells from the AF and NP, because most of the patients were of advanced age and their degenerated discs were removed piece by piece during spine surgery, making it difficult to precisely isolate cells from each location. In general, NP cells are more responsive to stimuli than AF cells [27]. Nevertheless, our findings observed in a mixed cell population may represent the overall action of IVD tissue. Fourth, U0126, which we used to block the ERK pathway, is a MEK1/2 (an upstream kinase of ERK) inhibitor, and not a direct inhibitor of ERK. Therefore, further experiments using selective inhibitors of ERK phosphorylation are necessary in the future to confirm their role in ERK phosphorylation.

## 4. Materials and Methods

### 4.1. Cell Culture

Human IVD cells were obtained from patients with LCS who underwent surgery in our hospital. The patients’ age range was 42 to 85 years (mean: 64.7 ± 11.1 years; *n* = 9). The Pfirrmann grading scale was used to score disc degeneration from Grade 1 (nondegenerated disc) to Grade 5 (severely degenerated disc) [59], and the discs that were obtained were all Grades 3 or 4.

This study was approved by the Ethics Review Committee of our institution (study approval no.: 1554, 27 October 2010). Written informed consent was obtained from all patients before the collection of specimens.

Immediately after surgery, human IVD cells were isolated as described previously [27]. Briefly, a mixture of the nucleus pulposus and annulus fibrosus was minced and digested with 0.1% pronase (Sigma-Aldrich, St. Louis, MO, USA) for 1 to 2 h and then with 0.1% collagenase (Sigma-Aldrich) overnight with shaking at 37 °C. The cells were maintained in a monolayer in Dulbecco’s Modified Eagle Medium (Life Technologies Corporation, Carlsbad, CA, USA) containing 10% fetal bovine serum (Life Technologies Corporation), and 1% penicillin-streptomycin (Life Technologies Corporation) at 37 °C under 5% CO_2_. Cells up to the fourth passage were used for the following experiments.

Human IVD cells seeded as a monolayer on 12-well plates at a density of 5 × 10^4^ cells per well were grown until confluency. Confluent cells were serum starved overnight, preincubated with PGE_2_ or PGE_1_ (1 µM) (Cayman Chemical, Ann Arbor, MI, USA) for 30 min, and then stimulated with recombinant human IL-1β (10 ng/mL) (PeproTech, Rocky Hill, NJ, USA) for 15, 30, 45, and 60 min, or preincubated with a MEK1/2 (upstream kinase of ERK) inhibitor (U0126), JNK inhibitor (SP600125), or p38 inhibitor (SB203580) (Merck, Darmstadt, Germany) at various concentrations for 30 min and then stimulated with IL-1β (10 ng/mL) for 24 h.

### 4.2. Western Blotting

The phosphorylation of MAPKs (ERK, JNK, and p38) was analyzed by Western blotting. Briefly, cells treated with various reagents were lysed with radioimmunoprecipitation assay buffer (20 mM Tris HCl (pH 7.4), 150 mM NaCl, 5 mM ethylenediaminetetraacetic acid, 1% (*v/v*) Triton X-100, 0.1% (*w/v*) sodium dodecyl sulfate (SDS), 1% (*w/v*) sodium deoxycholate, and 0.5% (*v/v*) Igepal CA-630) in the presence of protease and a phosphatase inhibitors cocktail (Merck) on ice. Total protein concentrations of lysates were measured using the BCA Protein Assay Kit (ThermoFisher Scientific, Waltham, MA, USA) and aliquots (50 µg) of total protein were subjected to SDS polyacrylamide gel electrophoresis with a 10% acrylamide gel under reducing conditions. The proteins separated in the gel were electrotransferred onto a polyvinylidene difluoride membrane and immunoreacted with an antibody (1:1000) against phosphorylated-ERK (#9101, Cell Signaling Technology, Danvers, MA, USA), phosphorylated-JNK (#4668, Cell Signaling Technology), phosphorylated-p38 (#4511, Cell Signaling Technology), or tubulin (Sigma-Aldrich). After reaction with a horseradish peroxidase-conjugated anti-rabbit immunoglobulin G antibody (1:5000) (sc-2313, Santa Cruz Biotechnology, Santa Cruz, CA, USA), immunoreactive proteins on the membranes were visualized using the ECL Western Blot Detection System (Merck) and imaged using ChemiDoc XRS plus (Bio-Rad Laboratories, Hercules, CA, USA). The intensity of the bands was quantified using Image Lab (Bio-Rad Laboratories), and expressed as a relative value to that of tubulin used as a loading control.

### 4.3. Real-Time Reverse Transcription–Polymerase Chain Reaction

Total RNA was extracted from the IVD cells using Total RNA Kit 1 (Omega Bio-Tek, Norcross, GA, USA) according to the manufacturer’s instructions. Quantitative real-time PCR was performed using Rotor-Gene Q (Qiagen, Valencia, CA, USA). A validated primer pair for DUSP-1 (HA212233) was purchased from Takara (Shiga, Japan), and those for NGF (QT00001589) and glyceraldehyde 3-phosphate dehydrogenase (GAPDH) (QT01192646) were purchased from Qiagen. Expression levels of the genes of interest were calculated and expressed as the difference relative to the housekeeping gene GAPDH, using the delta-delta Ct (threshold cycles) method.

### 4.4. siRNA Transfection

DUSP-1 knockdown in human IVD cells was performed by siRNA transfection. Human IVD cells were transfected with a validated DUSP-1-specific siRNA (#4390824, sense: 5′-*CCACCACCGUGUUCAACUUtt*-3′ and antisense: 5′-*AAGUUGAACACGGUGGUGGtg*-3′, ThermoFisher Scientific) or with a negative control siRNA (i.e., a sequence that is not complementary to any known mRNA; #4390843, ThermoFisher Scientific) using Lipofectamine RNAiMAX (ThermoFisher Scientific), according the manufacturer’s instructions. Briefly, the cells were seeded at 8.0 × 10^4^ cells/35-mm dish, cultured for 24 h, and then transfected either with DUSP-1 siRNA (5 nM) or with a negative control siRNA (5 nM). After 24 h, the culture medium was changed to serum-free medium. After another 24 h of culture, the cell medium was again replaced with fresh serum-free medium and then stimulated with IL-1β for the indicated period.

Transfection efficiency was monitored by transfecting the cells with BLOCK-iT Fluorescent Oligo (ThermoFisher Scientific) labeled with FITC. Fluorescence and bright-field images of the same cells were digitally captured using an EVOS FL microscope (ThermoFisher Scientific) and were merged using built-in software of the device (ThermoFisher Scientific).

### 4.5. Statistical Analysis

Data were compared by one-way analysis of variance with the Dunnett multiple comparison test using Prism 7 software (GraphPad Software, San Diego, CA, USA). Differences between two groups were analyzed using the Student *t*-test. *p*-values less than 0.05 were considered to indicate a statistically significant difference between groups.

## 5. Conclusions

The inhibition of IL-1β-induced NGF expression by PGE_2_ and PGE_1_ was found to be a result of the suppression of the signaling pathways of MAPKs, caused by increased DUSP-1 expression. MAPKs and DUSP-1 play important roles in NGF expression in IVD cells, and DUSP-1 may be a new target molecule for the treatment of LBP.

## Figures and Tables

**Figure 1 ijms-23-00371-f001:**
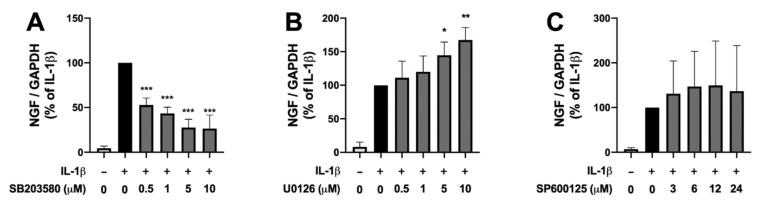
Involvement of various mitogen-activated protein kinase (MAPK) pathways in interleukin-1β (IL-1β)-induced nerve growth factor (NGF) expression in human intervertebral disc (IVD) cells. Confluent human IVD cells were serum starved, preincubated with the indicated concentrations of a p38 inhibitor (SB203580) (**A**), a MEK1/2 (upstream of extracellular signal-regulated kinase [ERK]) inhibitor (U0126) (**B**), or a c-Jun N-terminal kinase (JNK) inhibitor (SP600125) (**C**) for 30 min, and then stimulated with IL-1β (10 ng/mL) for 24 h. The relative expression levels of NGF were quantified by real-time PCR. Results are expressed as the mean ± SD (*n* = 4 individuals) after normalization to GAPDH, and expressed as a relative value to that of IL-1β alone. * *p* < 0.05, ** *p* < 0.01, and *** *p* < 0.001 vs. IL-1β alone.

**Figure 2 ijms-23-00371-f002:**
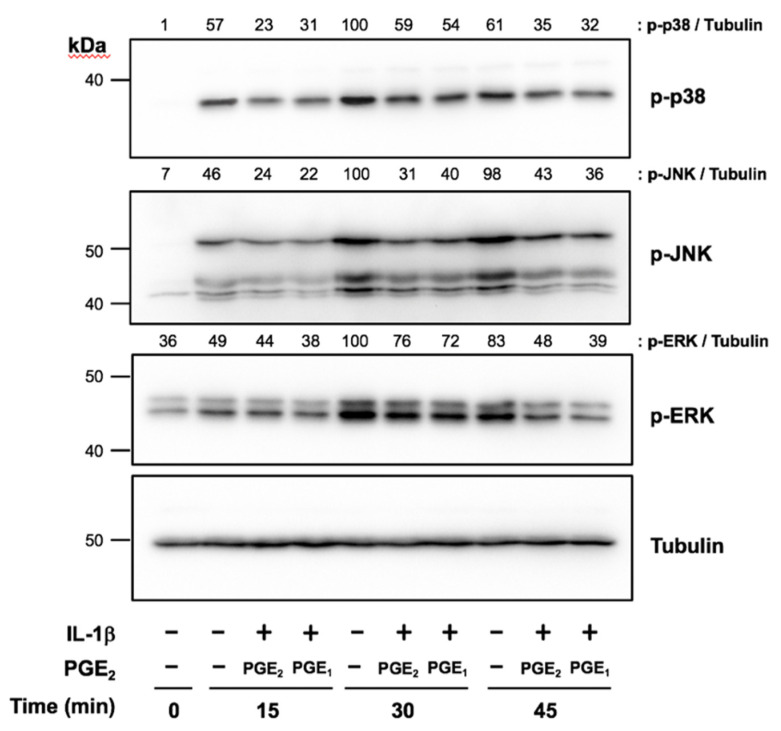
Effects of prostaglandin E_2_ (PGE_2_) and PGE_1_ on IL-1β-induced phosphorylation of MAPKs in human IVD cells. Confluent human IVD cells were serum starved, preincubated with PGE_2_ (1 μM) or PGE_1_ (1 μM) for 3 h, and then stimulated with IL-1β (10 ng/mL) for 15, 30, and 45 min. The phosphorylation of MAPKs was assessed by Western blotting. Results were reproducible among four individuals, and representative blots are shown.

**Figure 3 ijms-23-00371-f003:**
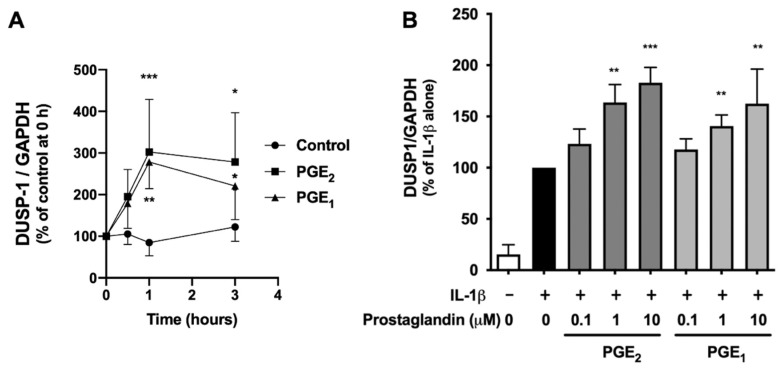
Effects of PGE_2_ and PGE_1_ on DUSP-1 expression in human IVD cells. (**A**) Confluent human IVD cells were serum starved, and then left untreated (circles), treated with PGE_2_ (1 μM) (squares), or treated with PGE_1_ (triangles) for 0.5, 1, and 3 h. Relative amounts of DUSP-1 were quantified by real-time PCR. Results are shown as the mean ± SD (*n* = 4 individuals) of DUSP-1 after normalization to GAPDH, and expressed as a relative value to that of untreated cells at each time point. * *p* < 0.05, ** *p* < 0.01, and *** *p* < 0.001 between untreated and PGE_2_- or PGE_1_-treated cells at each time point. (**B**) Confluent human IVD cells were serum starved, left untreated (black column), preincubated with PGE_2_ (light gray column), or preincubated with PGE_1_ (dark gray column) for 3 h (1, 10, or 100 µM), and then stimulated with IL-1β (10 ng/mL) for a further 6 h. Relative amounts of DUSP-1 were quantified by real-time PCR. Results are expressed as the mean ± SD (*n* = 4 individuals) of DUSP-1 after normalization to GAPDH, and expressed as a relative value to that of untreated cells at each time point. ** *p* < 0.01 and *** *p* < 0.001 between untreated and PGE_2_ or PGE_1_ treated cells at each time point.

**Figure 4 ijms-23-00371-f004:**
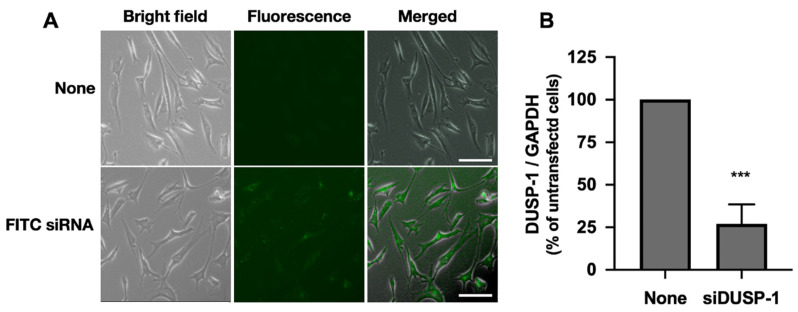
DUSP-1 knockdown by the transfection of small interfering RNA (siRNA) into human IVD cells. (**A**) Transfection efficiency of siRNA in human IVD cells. Semiconfluent human IVD cells were transfected with FITC-labeled siRNA oligonucleotides and cultured for 24 h. Cell images were captured by a digital fluorescence microscope. Transfection efficiency was more than 80% in three independent experiments (*n* = 3 individuals). Scale bar: 100 μm (**B**) DUSP-1 siRNA was transfected into semiconfluent human IVD cells to attenuate DUSP-1 expression. The relative amount of DUSP-1 expression was quantified by real-time PCR. Results are expressed as the mean ± SD (*n* = 3 individuals) of DUSP-1 after normalization to that of GAPDH, and expressed as a relative value to that of untransfected cells. DUSP-1 expression was significantly suppressed by more than 70%. *** *p* < 0.001 vs. untransfected cells.

**Figure 5 ijms-23-00371-f005:**
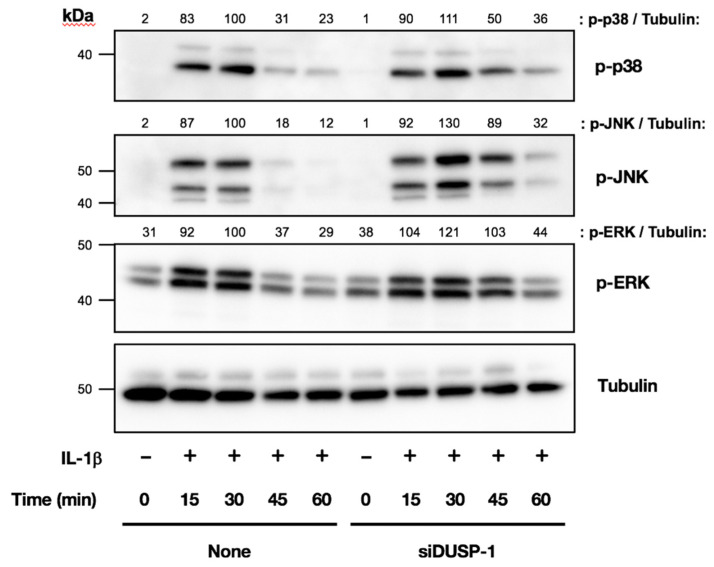
Effects of DUSP-1 knockdown on IL-1β-induced MAPK phosphorylation in human IVD cells. DUSP-1 knockdown and untransfected cells were serum starved and then stimulated with IL-1β (10 ng/mL) for 15, 30, 45, and 60 min. The phosphorylation of MAPKs was analyzed by Western blotting. IL-1β-induced MAPK phosphorylation was increased and prolonged in DUSP-1 knockdown cells compared with untransfected cells. Results were reproducible among three individuals and representative blots are shown.

**Figure 6 ijms-23-00371-f006:**
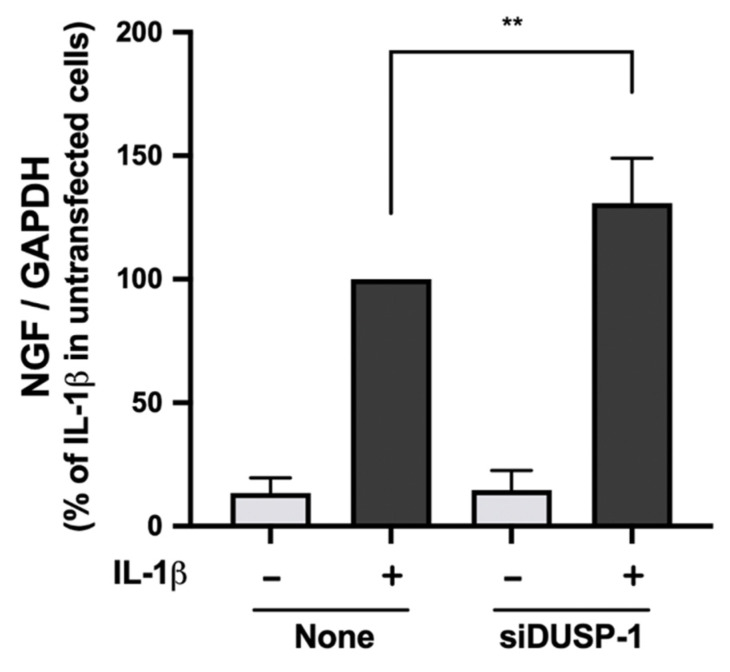
Effects of DUSP-1 knockdown on IL-1β-induced expression of NGF in human IVD cells. Untransfected and DUSP-1 knockdown human IVD cells were serum starved and stimulated with IL-1β (10 ng/mL) for 24 h. Expression levels of NGF were quantified by real-time PCR. Results are expressed as the mean ± SD (*n* = 4 individuals) after normalization to GAPDH, and are shown as relative values to that of untransfected cells stimulated with IL-1β. IL-1β-induced expression of NGF was significantly enhanced in DUSP-1 knockdown cells. ** *p* < 0.01 vs. untransfected cells stimulated with IL-1β.

## Data Availability

The data presented in this study are available from the corresponding author on reasonable request.

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
