# Peer review of "DUSP-1 Induced by PGE2 and PGE1 Attenuates IL-1β-Activated MAPK Signaling, Leading to Suppression of NGF Expression in Human Intervertebral Disc Cells"

_ijms, 2021, doi:10.3390/ijms23010371_

Round 1

Reviewer 1 Report

The study by Kusakabe and colleagues investigated the role of dual-specificity phosphatase (DUSP)-1 on the mechanism of IL-1β-induced NGF suppression by PGE2 and PGE1. The authors also focused on mitogen-activated protein kinases (MAPKs) pathway involved in inflammatory diseases. DUSP-1 suppressed the phosphorylation of MAPKs and may be a new therapeutic target for intervertebral disc degeneration. were The results showed that the use of different types of MAPK inhibitors changed the relative expression levels of NGF quantified by real-time PCR.  IVD cells were treated with PGE2 and PGE1 at different time points. Effects of PGE2 and PGE1 on IL-1β-induced phosphorylation of MAPKs were assessed by Western blotting.  DUSP-1 expression amounts was enhanced and IL-1β-induced phosphorylation of MAPKs was suppressed.  Disc cells were transfected with small interfering RNA (siRNA). IL-1β-induced MAPK phosphorylation and expression levels of NGF were increased in DUSP-1 knock-down cells compared with untransfected cells.

Major revisions should be required at this stage.

Author Response

We greatly appreciate the helpful and constructive comments. We have revised the manuscript in accordance with all of the comments raised. We hope that we have satisfactorily addressed all of the concerns, and that the revised manuscript is now suitable for publication in International Journal of Molecular Sciences. Our point-by-point responses to the comments raised are provided below.

Reviewer 1

Response:

Thank you very much for reviewing our manuscript. Unfortunately, we were not sure as to which part of the manuscript you were concerned about, and were not able to make any specific changes in accordance with your comments. However, we made several changes and improvements in accordance with the comments of the other 2 reviewers. We hope that out manuscript is now acceptable for publication in IJMS.

Reviewer 2 Report

U0126 is a selective inhibitor of MEK 1/2, but it blocks ERK phosphorylation. Mention ‘studies with a selective inhibitor of ERK 1/2 is required to a clearer conclusion on ERK phosphorylation. Discuss the previous papers with FR180204, which selectively inhibits ERK phosphorylation. Necessary changes to be made in the manuscript text and figure legends.

Figure 2 and Figure 5: Major concerns on the images. The immunoblots of pJNK show three bands. The pJNK has a molecular weight of 46/54. No bands above molecular weight 50 were seen. Whether all these blots are from a single membrane? If not, a separate housekeeping gene is needed. Figure legends state four independent experiments were conducted, each of MAPK with separate housekeeping genes needs to be represented. The authors missed looking at the total-p38, t-JNK, and t-ERK expressions, that is normally been shown with phosphorylated form. Also note that the antibody used to detect pJNK (#4668, CST) cross-react with p-p38 and p-ERK.

Introduction (line 75): These are not major classes of MAPK. MAPKs are of two types-typical and non-typical. The three MAPK mentioned come under typical. Correct the statement.

Line 93: ‘subtypes’ is not correct.

Figure 4B and Figure 6: The data needs to be presented in fold times (the expression of uninfected cells to be 1 and normalize to housekeeping gene)

More details on siRNA sequences are needed: The final concentration of siRNA used to be mentioned. The sequence of the siRNA designs including siControl needs to be provided in the manuscript.

Author Response

We greatly appreciate the helpful and constructive comments. We have revised the manuscript in accordance with all of the comments raised. We hope that we have satisfactorily addressed all of the concerns, and that the revised manuscript is now suitable for publication in International Journal of Molecular Sciences. Our point-by-point responses to the comments raised are provided below.

Reviewer 2

U0126 is a selective inhibitor of MEK 1/2, but it blocks ERK phosphorylation. Mention ‘studies with a selective inhibitor of ERK 1/2 is required to a clearer conclusion on ERK phosphorylation. Discuss the previous papers with FR180204, which selectively inhibits ERK phosphorylation. Necessary changes to be made in the manuscript text and figure legends.

Response:

Thank you very much for your important comments. It is true that U0126 is an inhibitor of MEK1/2, which are upstream kinases of ERK1/2, but is not a direct ERK1/2 inhibitor. We therefore changed our descriptions throughout the manuscript regarding U0126.

We routinely used U0126 to inhibit the MEK1/2-ERK1/2 pathway, because to our knowledge, there were no inhibitors specific for ERK available when we started our experiments. U0126 has been widely used in many studies and its specificity has been well characterized (please refer to Bain, J et al., Biochem J. [2007], 408, 297-315). On the other hand, FR180204 appears to be a relatively new inhibitor, and its specificity has not yet been confirmed by third parties. Therefore, although FR180204 appears to be a potentially useful inhibitor, we would like to use it in the future after it has been characterized more fully. We would hence like to leave it out of this manuscript, and ask for your kind understanding on this matter.

Figure 2 and Figure 5: Major concerns on the images. The immunoblots of pJNK show three bands. The pJNK has a molecular weight of 46/54. No bands above molecular weight 50 were seen. Whether all these blots are from a single membrane? If not, a separate housekeeping gene is needed. Figure legends state four independent experiments were conducted, each of MAPK with separate housekeeping genes needs to be represented. The authors missed looking at the total-p38, t-JNK, and t-ERK expressions, that is normally been shown with phosphorylated form. Also note that the antibody used to detect pJNK (#4668, CST) cross-react with p-p38 and p- ERK.

Response:

We thank you for these very important points. Firstly, we performed at least 1 Western blot for each of the 4 individual donors, and representative blots were shown in the manuscript. These Western blots were performed on a single membrane that was sequentially reprobed with different antibodies. The anti-pJNK antibody was the first antibody to be probed on the membrane, and therefore we believe that none of these 3 bands are from other blots. The pattern of the representative bands of pJNK detected with the pJNK antibody that we used (#4668) appears to be different depending on the cell type used, as stated in the manufacturer’s datasheet and other published papers. At least in the human intervertebral disc cells that we used, 3 bands were reproducibly seen in our Western blots.

Secondly, the molecular weight markers for the pJNK blots were mislabeled in the original manuscript figures. We apologize for our careless mistake, which we have corrected in the revised manuscript.

Thirdly, we agree that total p38, ERK, and JNK are often shown together with their phosphorylated form. However, we checked many papers that were recently accepted in IJMS, and found that the blots for the total protein were not always shown, and other proteins, such as tubulin or GAPDH were used as housekeeping genes for comparisons among the lanes on the same membrane. We therefore used tubulin as a loading control in our figures, and ask for your kind understanding on this matter.

Introduction (line 75): These are not major classes of MAPK. MAPKs are of two types-typical and non-typical. The three MAPK mentioned come under typical. Correct the statement.

Line 93: ‘subtypes’ is not correct.

Response:

We thank you for your comment. We agree that MAP kinase can be divided into 2 types, i.e., typical (ERK1/2, JNK1/2/3, p38α/β/γ/δ, and ERK5) and nontypical (ERK3/4, NLK, ERK7) MAP kinases. We hence corrected the text in accordance with your comment.

Figure 4B and Figure 6: The data needs to be presented in fold times (the expression of uninfected cells to be 1 and normalize to housekeeping gene)

Response:

              We believe that the major comparisons readers will want to make are, for Figure 4B, how much percent knockdown was obtained, and for Figure 6, how much enhancement of IL-1-induced NGF expression was observed. From this viewpoint, we believe that our original data presentation is reasonable, and ask for your kind understanding on this matter.

More details on siRNA sequences are needed: The final concentration of siRNA used to be mentioned. The sequence of the siRNA designs including siControl needs to be provided in the manuscript.

Response:

              The final concentration of siRNA was 5 nM for both DUSP-1 and control siRNA. The sequences of the siRNAs for DUSP-1 were as follows (sense: 5'-CCACCACCGUGUUCAACUUtt-3' and antisense: 5'-AAGUUGAACACGGUGGUGGtg-3'). The sequence of the control siRNA was not provided by the company, so we included the full product number instead.

We added the above details regarding the siRNA protocols to the Materials and Methods section of the revised manuscript.

Reviewer 3 Report

Review of IJMS 1504983

This a well written MS that proposes a novel mechanism in IVD pathobiology. Their hypothesis is strongly supported by the data they present. The figures are well presented and figure legends informative as they should be. The English is well presented throughout.

I only have very minor changes to suggest to what is otherwise an excellent manuscript.

Ref 1 what about the authors ? you should add GBD 2016 Disease and Injury Incidence and Prevalence Collaborators: Theo Vos, Amanuel Alemu Abajobir, Kalkidan Hassen Abate et al. to this reference.

A few references that the authors may also be interested in are:-

Shen B, Melrose J, Ghosh P, Taylor F. Induction of matrix metalloproteinase-2 and -3 activity in ovine nucleus pulposus cells grown in three-dimensional agarose gel culture by interleukin-1beta: a potential pathway of disc degeneration. Eur Spine J. 2003 Feb;12(1):66-75. Which was awarded the ESJ Grammer Prize in 2003.

Melrose J, Roberts S, Smith S, Menage J, Ghosh P. Increased nerve and blood vessel ingrowth associated with proteoglycan depletion in an ovine anular lesion model of experimental disc degeneration. Spine (Phila Pa 1976). 2002 Jun 15;27(12):1278-85.

Melrose J, Ghosh P, Taylor TK, Latham J, Moore R. Topographical variation in the catabolism of aggrecan in an ovine annular lesion model of experimental disc degeneration. J Spinal Disord. 1997 Feb;10(1):55-67.

Melrose J, Shu C, Young C, Ho R, Smith MM, Young AA, Smith SS, Gooden B, Dart A, Podadera J, Appleyard RC, Little CB. Mechanical destabilization induced by controlled annular incision of the intervertebral disc dysregulates metalloproteinase expression and induces disc degeneration. Spine (Phila Pa 1976). 2012 Jan 1;37(1):18-25.

A point the authors do not make in their manuscript is that nerve ingrowth into a degenerate IVD only occurs after the space-filling properties of aggrecan proteoglycan are removed from the IVD by elevated levels of proteolytic activity. Proteases degrade aggrecan and this also results in a reduced disc height and impaired biomechanical competence in the degenerate IVD. The high density of CS side chains on aggrecan in healthy IVDs inhibit nerve outgrowth, thus the degradation of aggrecan in the IVD provides a permissive environment for the penetration of nerves into this tissue. In normal IVDs with normal aggrecan levels nerves are confined to the outermost lamella of the AF and do not penetrate into this tissue.

Degradation of the IVD can be induced experimentally with a controlled annular lesion that destabilizes the IVD and results in up-regulation in MMPs that degrade the aggrecan. The above references provide relevant information to cover this.

Line 54 “interleukin (IL)-1β, IL-6 and tumor necrosis factor a “ add (TNFa) here

Author Response

We greatly appreciate the helpful and constructive comments. We have revised the manuscript in accordance with all of the comments raised. We hope that we have satisfactorily addressed all of the concerns, and that the revised manuscript is now suitable for publication in International Journal of Molecular Sciences. Our point-by-point responses to the comments raised are provided below.

Reviewer 3

This a well written MS that proposes a novel mechanism in IVD pathobiology. Their hypothesis is strongly supported by the data they present. The figures are well presented and figure legends informative as they should be. The English is well presented throughout.

I only have very minor changes to suggest to what is otherwise an excellent manuscript.

Ref 1 what about the authors ? you should add GBD 2016 Disease and Injury Incidence and Prevalence Collaborators: Theo Vos, Amanuel Alemu
Abajobir, Kalkidan Hassen Abate et al. to this reference.

Response:

              We thank you for pointing out this out. We corrected the format of reference.

A few references that the authors may also be interested in are:-

Shen B, Melrose J, Ghosh P, Taylor F. Induction of matrix metalloproteinase-2 and -3 activity in ovine nucleus pulposus cells grown in three-dimensional agarose gel culture by interleukin-1beta: a potential pathway of disc degeneration. Eur Spine J. 2003 Feb;12(1):66-75. Which was awarded the ESJ Grammer Prize in 2003.

Melrose J, Roberts S, Smith S, Menage J, Ghosh P. Increased nerve and blood vessel ingrowth associated with proteoglycan depletion in an ovine anular lesion model of experimental disc degeneration. Spine (Phila Pa 1976). 2002 Jun 15;27(12):1278-85.

Melrose J, Ghosh P, Taylor TK, Latham J, Moore R. Topographical variation in the catabolism of aggrecan in an ovine annular lesion model of experimental disc degeneration. J Spinal Disord. 1997 Feb;10(1):55-67.

Melrose J, Shu C, Young C, Ho R, Smith MM, Young AA, Smith SS, Gooden B, Dart A, Podadera J, Appleyard RC, Little CB. Mechanical destabilization induced by controlled annular incision of the intervertebral disc dysregulates metalloproteinase expression and induces disc degeneration. Spine (Phila Pa 1976). 2012 Jan 1;37(1):18- 25.

A point the authors do not make in their manuscript is that nerve ingrowth into a degenerate IVD only occurs after the space-filling properties of aggrecan proteoglycan are removed from the IVD by elevated levels of proteolytic activity. Proteases degrade aggrecan and this also results in a reduced disc height and impaired biomechanical competence in the degenerate IVD. The high density of CS side chains on aggrecan in healthy IVDs inhibit nerve outgrowth, thus the degradation of aggrecan in the IVD provides a permissive environment for the penetration of nerves into this tissue. In normal IVDs with normal aggrecan levels nerves are confined to the outermost lamella of the AF and do not penetrate into this tissue.

Degradation of the IVD can be induced experimentally with a controlled annular lesion that destabilizes the IVD and results in up-regulation in MMPs that degrade the aggrecan. The above references provide relevant information to cover this.

Response:

We thank you for your constructive comments that have helped us to improve our manuscript. In accordance with your comments, we added a more detailed description of the pathology of discogenic pain to the Introduction section of the revised manuscript and references.

Line 54 “interleukin (IL)-1β, IL-6 and tumor necrosis factor a “ add (TNFa) here

Response:

Tumor necrosis factor only appears on line 54, and is not used subsequently, so we believe that its abbreviation is not necessary.

Round 2

Reviewer 1 Report

The “Introduction” is well written. A sufficient overview on the matter is given.

 “Materials and Methods” section is sufficient described. Statistical analysis should be expressed as values ± SD and not SEM.

 “Results” should be improved:

  • The authors should normalize the phosphorylate proteins on relative non-phosphorylate proteins and not tubulin. Moreover the strip of tubulin could be repeated because it is not clear and used to normalize the ratio between phosphorylate and non phosphorylate proteins.
  • Please improve the fig.3 and show the DUSP1 silencing by western blotting too.

The “Discussion” is well written. The authors revised literature and discussed about  their results.

Figure legends: are well written.

Author Response

We greatly appreciate the helpful and constructive comments. We have re-revised the manuscript in accordance with all of the comments raised. We hope that we have satisfactorily addressed all of the concerns, and that the re-revised manuscript is now suitable for publication in International Journal of Molecular Sciences. Our point-by-point responses to the comments raised are provided below.

Reviewer 1

The “Introduction” is well written. A sufficient overview on the matter is given.

Response:

We thank you for your positive comment.

 “Materials and Methods” section is sufficient described. Statistical analysis should be expressed as values ± SD and not SEM.

Response:

We thank you for your comment. We changed all the bar graphs to show means ± SD rather than SEM. The text was also amended as appropriate.

 “Results” should be improved:

The authors should normalize the phosphorylate proteins on relative non-phosphorylate proteins and not tubulin. Moreover the strip of tubulin could be repeated because it is not clear and used to normalize the ratio between phosphorylate and non phosphorylate proteins.

Response:

We completely agree that it is rational to normalize the intensity of the phosphorylated protein to that of the total protein. However, unfortunately, we have had difficulty in stripping the antigen-antibody complex from the membrane using stripping reagents (we tried both “Re-blot Plus Strong Solution, Millipore, #2504” and the classical detergent-beta-mercaptoethanol-based stripping buffer). We tried to first probe the membrane with the anti-phospho antibody and then the anti-total antibody; however, we could not strip off the primary anti-phospho antibody completely (although the secondary antibody appeared to be stripped), and thisappeared to preventthe anti-total antibody from reacting with the antigen (the antigen appeared to be masked with the primary anti-phospho antibody), resulting in inconsistent results for the total protein blots. We therefore used tubulin as an alternative loading control. Although this may not be scientifically rational, using tubulin as a loading control appears to be accepted in many respected journals, including IJMS. We would hence like to ask for your kind understanding on this matter.

Please improve the fig.3 and show the DUSP1 silencing by western blotting too.

Response:

We thank you for your comment. We improved Figure 3 in accordance with your suggestion, by expressing the results as values ± SD rather than SEM. We believe that Figure 3 now expresses the results appropriately, and will be understandable to the readers.

Regarding the DUSP-1 protein, we agree that confirmation of DUSP-1 knockdown at the protein level is important. We have been struggling to detect the DUSP-1 protein, although we have tried 4 commercially available antibodies (Abcam, #EPR18884, lot# GR239206-2; Abnova, PAB6062, lot# S2G2; Santa Cruz Biotechnology, #sc-1102, lot# H2312; Cell Signaling Technology, #35217, lot# 1); we found that none of them provided consistent results when blotting human intervertebral disc cell lysates (one of the antibodies from Cell Signaling Technology detected the DUSP-1 protein only in human synovial fibroblasts).

One of the reasons for this difficulty may be that the steady-state amount of the DUSP-1 protein is lower than the detection level by the antibodies, owing to the low stability of the DUSP-1 protein. The half-life of the DUSP-1 protein appears to be relatively short according to a previous study, in which the authors performed an S35-methionine pulse-chase experiment in CCL39 fibroblasts, and reported that the half-life of the DUSP-1 protein was 45 min (Brondello et al, Science1999, 286, pp 2514-7). It may be possible that the half-life of DUSP-1 may differ among cell types, and may hence be very short in human intervertebral disc cells.

Therefore, we clearly described this issue in the limitation part of the Discussion section,and ask for your kind understanding on this matter.

The “Discussion” is well written. The authors revised literature and discussed about their results.

Response:

We thank you for your positive evaluation.

Figure legends: are well written.

Response:

We thank you for your positive evaluation.

Reviewer 2 Report

Comment for Author Response 1: Even though the authors respond with their clarification on ERK1/2 phosphorylation with U0126. I am not convinced by their statements provided. A necessary statement in the manuscript is required with more modifications. " Even though experiments were conducted with U0126, more experiments with selective inhibitors of ERK1/2 phosphorylation is necessary to conclude their role on ERK1/2 phosphorylation"

Comments for Author Response 2: I partially agree with the authors. In the case of total proteins, I completely disagree with the author's response. The detection of total proteins gives more evidence-based information on your western bands, as both phosphorylated and total proteins have the same molecular weight. This makes more surety on your protein of interest with the individual band (in practice phospho has to be probed first and then the total to be probed). For scientific practice, the researchers should have their own aim to have clarity on their results, when presented to others rather than explaining with some reason, like what others are doing. If a mistake happens, and any other person follows the same, the same mistakes continue. I request the authors to think wisely and come to a conclusion to address this specific question.

Other responses provided by the authors are satisfactory.

Author Response

We greatly appreciate the helpful and constructive comments. We have re-revised the manuscript in accordance with all of the comments raised. We hope that we have satisfactorily addressed all of the concerns, and that the re-revised manuscript is now suitable for publication in International Journal of Molecular Sciences. Our point-by-point responses to the comments raised are provided below.

Reviewer 2

Comment for Author Response 1: Even though the authors respond with their clarification on ERK1/2 phosphorylation with U0126. I am not convinced by their statements provided. A necessary statement in the manuscript is required with more modifications. " Even though experiments were conducted with U0126, more experiments with selective inhibitors of ERK1/2 phosphorylation is necessary to conclude their role on ERK1/2 phosphorylation"

Response:

We thank you for your constructive comments. We agree that we cannot conclude the role of the ERK pathway on IL-1β-induced NGF expression from our experiments using U0126. This is clearly a limitation of this study. We therefore added the statement that you suggested as a limitation, to the Discussion section of the revised manuscript.

Comments for Author Response 2: I partially agree with the authors. In the case of total proteins, I completely disagree with the author's response. The detection of total proteins gives more evidence-based information on your western bands, as both phosphorylated and total proteins have the same molecular weight. This makes more surety on your protein of interest with the individual band (in practice phospho has to be probed first and then the total to be probed). For scientific practice, the researchers should have their own aim to have clarity on their results, when presented to others rather than explaining with some reason, like what others are doing. If a mistake happens, and any other person follows the same, the same mistakes continue. I request the authors to think wisely and come to a conclusion to address this specific question.

Response:

We completely agree that presentation of both phosphorylated and total proteins for the kinase experiments would provide more evidence-based information, and that researchers should have their own aim to have clarity regarding their results. We would also like to apologize for our previous response to your comment, which was scientifically inappropriate.

The scientific aim of showing both phosphorylated and total protein by Western blotting, we believe, is to exclude the possibility that the changes in the amount of phosphorylated form are not owing to changes in the amount of their total protein among the blots to be compared. We agree that the best way is to show blots of both phosphorylated and total protein from the same membrane; however, we have had difficulty in stripping the antigen-antibody complex from the membrane using stripping reagents (we tried both “Re-blot Plus Strong Solution, Millipore, #2504” and the classical detergent-beta-mercaptoethanol-based stripping buffer). We tried to first probe the membrane with the anti-phospho antibody and then the anti-total antibody; however, we could not strip off the primary anti-phospho antibody completely (although the secondary antibody appeared to be stripped), and this appeared to prevent the anti-total antibody from reacting with the antigen (the antigen appeared to be masked with the primary anti-phospho antibody), resulting in inconsistent results for the total protein blots. We therefore used tubulin as an alternative loading control. We would hence like to ask for your kind understanding on this matter.

Other responses provided by the authors are satisfactory.

Response:

We thank you for your positive evaluation.

Round 3

Reviewer 1 Report

The authors reported the corrections as requested. The paper is suitable to be  accepted.

Reviewer 2 Report

The comments provided are satisfactory. Hence, the manuscript is recommended for acceptance.